# School Lunch and Body Size in Japanese Junior High School Students: The Japanese National Health and Nutrition Survey

**DOI:** 10.3390/nu17050895

**Published:** 2025-03-03

**Authors:** Suzuna Iwano, Kotone Tanaka, Aru Takaoka, Daisuke Machida, Yasutake Tomata

**Affiliations:** 1School of Nutrition and Dietetics, Faculty of Health and Social Services, Kanagawa University of Human Services, Yokosuka 238-8522, Japan; 2Department of Home Economics Education, Cooperative Faculty of Education, Gunma University, Maebashi 371-8510, Japan

**Keywords:** body mass index, child, energy intake, junior high school student, normal weight, obesity, school lunch, underweight

## Abstract

**Objectives**: Although the school lunch program is expected to reduce obesity and underweight among children in Japan, there had been no individual-level study examining the impact of school lunch on body size (overweight or underweight). The present study examined the association between school lunch and body size in Japanese junior high school students. **Methods**: This cross-sectional study was conducted based on data from the Japanese National Health and Nutrition Survey in 2014 and 2018. The present analysis included 323 individuals (12–15 years old). The exposure factor was school lunch usage. The primary outcome measure was body size (normal weight, overweight/obesity (including both overweight and obesity), and underweight). **Results**: Of 323 individuals, the proportion of school lunch users was 65.6%. School lunch was not statistically significantly associated with normal weight; the multivariate-adjusted odds ratio (95% confidence interval) of normal weight in school lunch users was 1.07 (0.66–1.75) in comparison with non-users. No significant associations were found for overweight/obesity or underweight outcomes. **Conclusions**: The present findings did not support the expectation for the Japanese school lunch program.

## 1. Introduction

Childhood obesity increases the risk of obesity in adulthood and increases the risk of chronic diseases such as cardiovascular disease and mortality in young adulthood [1,2,3]. Additionally, underweight in childhood has been identified as a risk factor for bone fractures in later adulthood [4]. Therefore, maintaining normal weight through a proper diet from childhood is expected to contribute to better health in adulthood. The national health promotion program, “The Health Japan 21 (the third term)”, includes reducing overweight and obesity in children and underweight in young women as a goal based on a life-course perspective [5,6].

Japan has a national program of school lunch under the law “the School Lunch Program Act” [7]. The percentage of Japanese children who are overweight or obese is relatively low compared to other high-income countries. School lunches have been assumed as constituting a potential reason for this low rate [8,9]. The School Lunch Program Act states that the purpose of school lunches is to “contribute to the sound physical and mental development of children and students”. The Japanese school lunch is provided in accordance with “the School Lunch Nutrition Standards” [10,11,12]. Previous studies had indicated higher intakes of protein, dietary fiber, and micronutrients as benefits of school lunches [13,14]. Such a nationwide school lunch program is a unique feature of Japan and is expected to be one of the factors contributing to Japan’s lower prevalence of obesity. One Japanese ecological study (prefectural-level) reported that higher school lunch coverage in junior high school students was associated with lower percentages of those who were overweight or obese, especially among boys [9]. However, this previous study reported no difference in the percentages of underweight students or obese girls [9]. To our knowledge, no studies had examined these associations using individual-level data in Japan.

Outside of Japan, have been there are several research reports that have examined these associations using individual-level data. One Korean study reported that elementary through high school students without school lunches tended to have lower odds ratios for obesity than those with school lunches [15]. The other Korean study reported that the body mass index (BMI) decreased in areas where school lunches were discontinued compared to areas where school lunches were continued and that the BMI increased with the resumption of school lunches [16]. In the U.S. studies, there are have been mixed reports of both increased and decreased rates of obesity among those who have participated in school lunch programs, with no unanimous conclusions [17,18,19,20,21]. In summary, studies on the association between school lunches and obesity have had inconsistent conclusions. Furthermore, to our knowledge, the impact of school lunch on childhood underweight in high–income countries has not yet been examined.

Therefore, it would be useful to examine the association between school lunches and body size in the Japanese population using individual-level epidemiological data. The purpose of this study was to examine the relationship between school lunches and body size among junior high school students.

## 2. Materials and Methods

### 2.1. Study Design

This study was a cross-sectional study using pooled data from the National Health and Nutrition Survey in 2014 and 2018 [22,23]. The survey population consisted of all households in randomly selected unit-districts [24,25]. Unit-districts were defined based on address. In each survey, 300 unit-districts were randomly selected in two steps from the main residential areas (excluding unit-districts predominantly containing forests and public facilities) of all unit-districts nationwide across Japan.

The numbers of households surveyed were 5432 in the 2014 survey and 5032 in the 2018 survey. In the 2014 survey, 3648 out of 5432 households participated, while in the 2018 survey, 3268 out of 5032 households participated [24,25].

### 2.2. Study Participants

Figure 1 shows the flow diagram of study participants. Of the eligible household members who responded, 456 individuals (2014: *n* = 267; 2018: *n* = 189) were aged 12–15 years and were “junior high school students”. From these 456 individuals, those with missing energy intake (*n* = 13) or weight data (*n* = 120) were excluded, leaving 323 individuals for analysis in the present study.

### 2.3. Exposure Factor

The exposure factor was the intake of school lunch. Those who responded “school lunch” on the “Nutritional Intake Status Survey” were defined as individuals with school lunch (“user”), and the others were defined as those without school lunch (“non-user”) [22].

### 2.4. Outcomes

The primary outcome measure was body size (“normal weight”, “overweight/obesity”, and “underweight”). Furthermore, energy intake was used as a secondary outcome.

BMI was calculated using both self–reported and measured height and weight. The International Obesity Task Force (IOTF) BMI reference values were used to determine overweight, obese, and underweight participants [26]. The IOTF BMI reference values are the BMI values for each month of age, which correspond to the BMI at 18 years of age, separately for males and females. In this study, values of 144 months were used for individuals aged 12, 156 months for individuals aged 13, 168 months for individuals aged 14, and 180 months for individuals aged 15. Three outcome variables were defined: normal weight, overweight/obesity (including both overweight and obesity; ≥overweight), and underweight. All three outcome variables were binary variables (e.g., normal weight or not).

Energy intake was calculated from the dietary record data obtained from the responses to “Nutritional Intake Status Survey” [22]. For the school lunches, the list of foods used for meal planning was obtained and participants were asked for leftover of the school lunches, and nutritionist researchers calculated their intake [27].

### 2.5. Covariates

Adjustment variables were age, gender, annual household income, and number of household members. Household income was obtained using “Lifestyle Habits Questionnaire” [22], with four groups in the 2014 National Health and Nutrition Survey (<2 million JPY, 2–<6 million JPY, ≥6 million JPY, Missing) and five groups in the 2018 National Health and Nutrition Survey (<2 million JPY, 2–<4 million JPY, 4–<6 million JPY, ≥6 million JPY, Missing). Four household income groups were defined for the present analysis: “<2 million JPY”, “2–<6 million JPY” (including “2–<4 million JPY” and “4–<6 million JPY” in 2018), “≥6 million JPY”, and “missing”. The number of household members was derived from the “Household Status” section of the “the Nutritional Intake Status Survey” [22].

### 2.6. Ethical Considerations

The present study was conducted with the approval of the Research Ethics Committee of Kanagawa University of Human Services. The National Health and Nutrition Examination Survey is conducted by the Japanese government under the Japanese law “Health Promotion Act”, and research utilization of Japanese government statistics is stipulated in the “Statistics Act”.

### 2.7. Statistical Analyses

As the main analysis, binomial logistic regression analysis was used to calculate multivariate adjusted odds ratios and 95% confidence intervals for body size using the no-school-lunch group as the reference. Adjustment items were gender, age, household income, and number of household members.

Three types of sensitivity analyses were also performed. First, because the main analysis included overweight to account for the situation that obesity is relatively rare in the Japanese population, we performed an analysis with obesity (not including overweight) as the outcome variable. Second, to check whether the results for each gender were consistent with the main results, a stratified analysis by gender was performed. Third, to examine whether the impact of school lunch on body size differed depending on household income (e.g., positive association between school lunch and normal weight was stronger for students with lower household income), a stratified analysis by household income was performed.

In addition, a general linear model was used to calculate adjusted means and 95% confidence intervals of energy intake by school lunch group. In addition, adjusted means and 95% confidence intervals of energy intake by school lunch group were also calculated when stratified by body size (normal weight, overweight/obesity, and underweight). The reason for conducting this stratified analysis was to consider whether energy intake was appropriate for body size—for example, whether obese individuals who were school lunch users had relatively lower energy intake than non-users. As a sensitivity analysis of energy intake, a stratified analysis by gender was also performed.

Statistical significance was set at *p* < 0.05. All statistical analyses were performed using R ver. 4.3.1.

## 3. Results

### 3.1. Basic Characteristics

Of the 323 individuals analyzed, 65.6% were the “user” group of school lunch. The distribution of body size was as follows: 64.7% normal weight, 26.9% overweight/obesity, and 8.4% underweight. Table 1 shows the basic characteristics of the individuals according to their school lunch groups. The group with school lunch tended to have lower household incomes.

### 3.2. School Lunch and Body Size

As a result of the main analysis, Table 2 shows adjusted the odds ratios (95% confidence intervals) of body size (normal weight, overweight/obesity, and underweight) for the school lunch group. There was no statistically significant association between school lunch and normal weight; odds ratios of normal weight for the “user” group were 1.12 (0.69–1.81) in Model 1 and 1.07 (0.66–1.75) in Model 2. There was no statistically significant association between school lunch and overweight/obesity; odds ratios of overweight/obesity for the “user” group were 0.98 (0.58–1.65) in Model 1 and 0.99 (0.58–1.69) in Model 2. There was no statistically significant association between school lunch and underweight; odds ratios of underweight for the “user” group were 0.75 (0.33–1.68) in Model 1 and 0.86 (0.38–1.97) in Model 2.

The results of the stratified analysis by gender are also shown in Table 2. Similar to the results for all individuals (*n* = 323), there were no significant associations; odds ratios (95% confidence interval) of normal weight in the “user” group were 1.06 (0.54–2.09) for boys and 1.10 (0.53–2.27) for girls in Model 2. However, the directions of the odds ratios (point estimates) when looking at outcomes overweight/obesity and underweight differed in boys and girls; odds ratios (Model 2) of overweight/obesity in the “user” group were 0.88 for boys and 1.15 for girls and odds ratios (Model 2) of underweight in the “user” group were 1.21 for boys and 0.64 for girls.

Table 3 shows the results of the sensitivity analysis for obesity. There was no statistically significant association between school lunch and obesity; the odds ratios (95% confidence interval) were 1.78 (0.48–6.68) in Model 1 and 1.99 (0.52–7.62) in Model 2.

Table 4 shows the results of the analysis stratified by household income. There were no significant associations; the odds ratios (95% confidence intervals) of normal weight in the “user” group were 0.25 (0.15–4.12) for “<2 million JPY”, 1.57 (0.71–3.45) for “2–<6 million JPY”, 0.87 (0.43–1.75) for “≥6 million JPY”, and 1.55 (0.06–39.62) for “missing” in Model 2.

### 3.3. School Lunch and Energy Intake

Table 5 shows adjusted means (95% confidence interval) of energy intake. There was no statistically significant difference in energy intake between the school lunch groups: 2238 (2173–2389) kcal for “user” and 2235 (2103–2366) kcal for “non-user”, respectively. When the results were stratified by body size (normal weight, overweight/obesity, and underweight), there were also no significant differences in the mean energy intake for either group.

By gender, although the boys in the “user” group of school lunch had 40 kcal lower than those in “non-user”, the girls in the “user” group had 152 kcal higher than those in “non-user”.

## 4. Discussion

This cross-sectional study, which used national individual-level data in Japan, examined the association between school lunch and body size among junior high school students. As a main result, there was no statistically significant association between school lunch and body size.

A previous study had shown that school lunch was negatively associated with overweight/obesity, especially among boys [9]. On the other hand, in the present study, the association was not statistically significant, regardless of gender. However, since this previous study was an ecological study using prefecture-level data (47 prefectures in Japan) [9], a simple comparison may not be appropriate. Future research at the individual level will be required to obtain integrative evidence, such as via meta-analysis.

In the present study, the point estimate (the adjusted odds ratio) of underweight was lower than that of overweight/obesity in school lunch users, which can be interpreted as a relatively stronger association, although all of these results were not statistically significant. In addition, this tendency was relatively pronounced for girls, although all of these results were not statistically significant. One potential explanation for this tendency might be the influence of body image concerns and the desire to be thin. A previous study had shown that girls are more likely to desire thinness [28], and it is conceivable that in the absence of school lunch (i.e., when students bring their own lunches or purchase food independently), girls with a stronger desire to be thin may engage in restrictive eating behaviors, leading to an increased risk of underweight. In contrast, school lunches in Japan are provided in accordance with the School Lunch Nutrition Standards and are uniformly distributed regardless of students’ individual weight-loss intentions. This standardization may help ensure that students, even those with a strong desire to lose weight or who are underweight, consume an adequate amount of food in schools. Supporting this speculation, higher energy intake in the school lunch users compared to the non-users was observed, among girls across all body size strata. Because the previous study had suggested that the school lunch program reduces income disparities in micronutrient insufficiency among elementary school students [14], the school lunch program may have a favorable role in addressing nutritional insufficiency relatively. However, since none of the results in the present study were statistically significant, the above speculations should be confirmed by future studies. In addition, the previous study reported that the median (interquartile range) percentage of leftover food on a weight basis for junior high school lunches was 9.3 (4.6, 16.0) % [29], indicating that it was common for considerable leftovers to occur. Therefore, it would not be appropriate to interpret the findings on the assumption that students were consuming the entire school lunch provided.

The differences in mean energy intake between the school lunch groups may explain the contribution of school lunch to body size. The difference in the mean of energy intake between the school lunch groups (users–non-users) was −265 kcal for boys who were overweight/obese and 213 kcal for girls who were overweight/obese and 295 kcal for those who were underweight. These differences approximated the energy deficit or surplus required for a 1 kg change in body fat over one month (approximately 230 kcal/day) [30]. These results implied that the current school lunch system, which provides uniform meals, may not be able to optimize the energy intake of overweight/obese and underweight students (i.e., control the excessive energy intake of overweight/obese students and satisfy the energy intake of underweight students). In order to optimize the energy intakes of all students and maintain or improve their nutritional statuses, it is desirable to set energy intake standards that take into account the physique and living conditions of each individual student and to take individualized measures such as adjusting the amount of staple food intake.

Although we attempted to examine the impact of low household income (e.g., <2 million JPY versus ≥2 million JPY), it was difficult to calculate reliable estimations since only 19 participants had a household income of <2 million JPY. Whether the impact of school lunches on body size varies by household income could not be clarified by the present study.

There are both advantages and disadvantages to implementing the uniform school lunch approach such as those noted in the program in Japan. As an advantage, it has been reported that the school lunch may reduce the gap in micronutrient intake even among children from different socioeconomic backgrounds, as aforementioned [14]. These expectations have been suggested worldwide [31,32,33,34]. The Japanese school lunch is a nutrition policy that is equally accessible to all children [33]. The current Japanese school lunch program has a historical background, having been started for students who experienced starvation after World War II, and nutrient sufficiency has been a target of the school lunch program for more than half a century [12]. However, in Japan, since the 21st century, it has been reported that the frequency of child obesity is higher in households with lower economic statuses, and it is difficult to say that starvation is a more common public health issue even among households with lower economic statuses [35,36,37]. Additionally, the uniformity of the meal content emphasizes the educational value of teaching children about food and food culture in relation to nutrition [38]. On the other hand, one disadvantage is that the uniform school lunch approach may not address individual dietary needs and preferences. Preferences have been reported as factors associated with leftover school lunches in Japanese junior high schools [29]. Using the example of school lunches in Japan and France, the previous study had also suggested that there was a concern that education that encouraged uniform school lunches may result in children whose school lunch contents did not match their personal culture and preferences, thereby contributing to social exclusion [33,39]. Furthermore, for strategies to increase school lunch consumption, research evidence also supports offering students more menu choices and adapting recipes to improve the palatability and/or cultural appropriateness of foods [40]. A recent systematic review stated that “evidence shows voluntary policies are generally not an effective approach for shaping and protecting a healthy food environment” regarding the school lunch program in Japan [41]. This previous study concluded that “Future research should evaluate the implementation and efficacy of national school meal policies for improving health” [41].

Several limitations of this study should be considered. First, there was a possibility of underestimation due to the misclassification of school lunch exposure. The dietary survey conducted in the National Health and Nutrition Survey collects data from a single day (excluding Sundays and national holidays) [24,25]. A certain number of students in the non-users of school lunch may have continually used school lunches at times other than on the survey day (e.g., temporary non-use due to absence from school), potentially underestimating the effect of school lunch on body size. Unfortunately, we lacked data to confirm the extent of this misclassification. Sampling bias may have caused underestimation. Although the National Health and Nutrition Survey randomly sampled areas of the country, survey participation rates were not high (in 2014, for men: 48.7%; in 2014, for women: 51.3%; in 2018, for men: 45.4%; in 2018, for women: 47.6%) [42]. Therefore, the possibility that the results of the present study were underestimated due to sampling bias cannot be ruled out. Second, energy intake data might contain considerable measurement errors. The National Health and Nutrition Survey of Japan employs a “proportional distribution method” for dietary survey [43]. Regarding school lunches, we have not been able to obtain detailed information on the amount of food consumed. Furthermore, a previous study had reported that Japanese individuals may underreport their energy intake, particularly those with obesity (aged 1–19 years) [44]. Third, this study was a cross-sectional study. We cannot rule out the possibility of a reverse causation (e.g., that school lunches might be implemented because the area in question has a chronic high number of obese or thin students). Fourth, the overall sample size of 323 people was not large. In particular, the number of analysis subjects who were obese, the outcome of the sensitivity analysis, was even more limited, with three participants in the no-lunch group and ten participants in the with-lunch group. Furthermore, it was difficult to obtain robust results in the analysis stratified by household income. In a previous study of students of the same age as the present study in one region of Japan, the percentage of obesity was 3.0% for boys and 1.9% for girls [45]. Since the percentage of obese students in the present study was not significantly smaller compared to that in the Japanese previous study, it would be hard to conduct a statistical test using obesity alone as the outcome. However, the point estimate of the odds ratio for obesity as an outcome in the present study was 1.99. Therefore, based on the results of the present study, it would be difficult to observe that school lunches reduced obesity, even if the sample size were increased.

## 5. Conclusions

The findings of the present study did not support the expectation “Japanese school lunch program in junior high school contributes to an increase in the number of junior high school students with normal weight in Japan”. Additionally, the findings of the present study did not clarify whether the school lunch program contributed to providing an appropriately sized meal for each student. Further study will be required to implement a school lunch program that takes into account individualized nutrition management to provide an appropriate serving size for each student.

## Figures and Tables

**Figure 1 nutrients-17-00895-f001:**
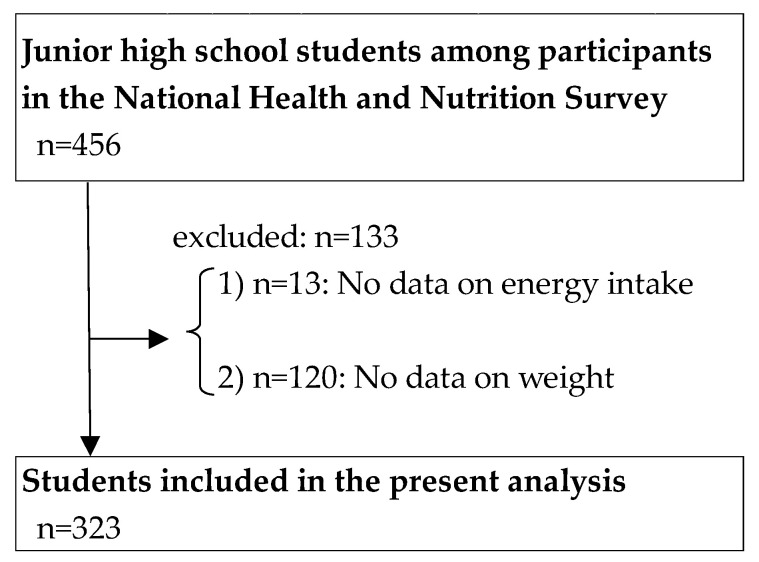
Flowchart of study participants.

**Table 1 nutrients-17-00895-t001:** Basic characteristics according to school lunch groups (*n* = 323).

	School Lunch	
	Non-User	User
Number of participants, *n*	111	212
Age (years), mean ± SD	13.7 ± 0.9	13.6 ± 1.0
Sex, *n* (%)		
Boys	59 (53.2)	114 (53.8)
Girls	52 (46.8)	98 (46.2)
Household income (Japanese JPY/year), *n* (%)		
<2 million JPY	5 (4.5)	14 (6.6)
2–<6 million JPY	40 (36.0)	92 (43.4)
≥6 million JPY	60 (54.1)	97 (45.8)
Missing	6 (5.4)	9 (4.2)
Number of household members, mean ± SD	4.2 ± 1.1	4.5 ± 1.2

**Table 2 nutrients-17-00895-t002:** Association between school lunch and body size (*n* = 323).

	School Lunch		*p*
	Non–User	User	
All			
Number of participants, *n*	111	212	
Normal weight			
Number of cases, *n* (%)	70 (63.1)	139 (65.6)	
Model 1 ^a^	1 (ref)	1.12 (0.69–1.81)	0.637
Model 2 ^b^	1 (ref)	1.07 (0.66–1.75)	0.775
Overweight/obesity			
Number of cases, *n* (%)	30 (27.0)	57 (26.9)	
Model 1 ^a^	1 (ref)	0.98 (0.58–1.65)	0.944
Model 2 ^b^	1 (ref)	0.99 (0.58–1.69)	0.970
Underweight			
Number of cases, *n* (%)	11 (9.9)	16 (7.5)	
Model 1 ^a^	1 (ref)	0.75 (0.33–1.68)	0.486
Model 2 ^b^	1 (ref)	0.86 (0.38–1.97)	0.719
Boys			
Number of participants, *n*	59	114	
Normal weight			
Number of cases, *n* (%)	38 (64.4)	76 (66.7)	
Model 1 ^c^	1 (ref)	1.10 (0.57–2.14)	0.768
Model 2 ^d^	1 (ref)	1.06 (0.54–2.09)	0.857
Overweight/obesity			
Number of cases, *n* (%)	17 (28.8)	30 (26.3)	
Model 1 ^c^	1 (ref)	0.88 (0.44–1.78)	0.728
Model 2 ^d^	1 (ref)	0.88 (0.43–1.80)	0.731
Underweight			
Number of cases, *n* (%)	4 (6.8)	8 (7.0)	
Model 1 ^c^	1 (ref)	1.04 (0.30–3.60)	0.954
Model 2 ^d^	1 (ref)	1.21 (0.33–4.36)	0.772
Girls			
Number of participants, *n*	52	98	
Normal weight			
Number of cases, *n* (%)	32 (61.5)	63 (64.3)	
Model 1 ^c^	1 (ref)	1.14 (0.57–2.29)	0.716
Model 2 ^d^	1 (ref)	1.10 (0.53–2.27)	0.800
Overweight/obesity			
Number of cases, *n* (%)	13 (25.0)	27 (27.6)	
Model 1 ^c^	1 (ref)	1.10 (0.50–2.38)	0.817
Model 2 ^d^	1 (ref)	1.15 (0.51–2.61)	0.737
Underweight			
Number of cases, *n* (%)	7 (13.5)	8 (8.2)	
Model 1 ^c^	1 (ref)	0.61 (0.21–1.81)	0.371
Model 2 ^d^	1 (ref)	0.64 (0.20–2.05)	0.449

^a^ Adjusted for age and sex. ^b^ Adjusted for age, sex, household income (<2 million JPY, 2–<6 million JPY, ≥6 million JPY, missing), and number of household members. ^c^ Adjusted for age. ^d^ Adjusted for age, household income (<2 million JPY, 2–<6 million JPY, ≥6 million JPY, missing), and number of household members.

**Table 3 nutrients-17-00895-t003:** Sensitivity analysis: school lunch and obesity (*n* = 323).

	School Lunch		*p*
	Non–User	User	
Number of participants, *n*	111	212	
Obesity			
Number of cases, *n* (%)	3 (2.7)	10 (4.7)	
Model 1 ^a^	1 (ref)	1.78 (0.48–6.68)	0.390
Model 2 ^b^	1 (ref)	1.99 (0.52–7.62)	0.314

^a^ Adjusted for age and sex. ^b^ Adjusted for age, sex, household income (<2 million JPY, 2–<6 million JPY, ≥6 million JPY, missing), and number of household members.

**Table 4 nutrients-17-00895-t004:** Analysis stratified by household income (*n* = 323).

	School Lunch		*p*
	Non-User	User	
<2 million JPY			
Number of participants, *n*	5	14	
Normal weight			
Number of cases, *n* (%)	4 (80.0)	7 (50.0)	
Model 1 ^a^	1 (ref)	0.27 (0.17–4.27)	0.354
Model 2 ^b^	1 (ref)	0.25 (0.15–4.12)	0.334
Overweight/obesity			
Number of cases, *n* (%)	1 (20.0)	7 (50.0)	
Model 1 ^a^	1 (ref)	3.69 (0.23–58.08)	0.354
Model 2 ^b^	1 (ref)	4.03 (0.24–68.26)	0.334
Underweight			
Number of cases, *n* (%)	0 (0.0)	0 (0.0)	
Model 1 ^a^	-	-	
Model 2 ^b^	-	-	
2–<6 million JPY			
Number of participants, *n*	40	92	
Normal weight			
Number of cases, *n* (%)	22 (55.0)	63 (68.5)	
Model 1 ^a^	1 (ref)	1.76 (0.82–3.79)	0.146
Model 2 ^b^	1 (ref)	1.57 (0.71–3.45)	0.263
Overweight/obesity			
Number of cases, *n* (%)	14 (35.0)	24 (26.1)	
Model 1 ^a^	1 (ref)	0.66 (0.29–1.46)	0.302
Model 2 ^b^	1 (ref)	0.72 (0.32–1.64)	0.436
Underweight			
Number of cases, *n* (%)	4 (10.0)	5 (5.4)	
Model 1 ^a^	1 (ref)	0.53 (0.13–2.12)	0.370
Model 2 ^b^	1 (ref)	0.61 (0.14–2.55)	0.495
≥6 million JPY			
Number of participants, *n*	60	97	
Normal weight			
Number of cases, *n* (%)	40 (66.7)	62 (63.9)	
Model 1 ^a^	1 (ref)	0.90 (0.45–1.78)	0.759
Model 2 ^b^	1 (ref)	0.87 (0.43–1.75)	0.700
Overweight/obesity			
Number of cases, *n* (%)	14 (23.3)	25 (25.8)	
Model 1 ^a^	1 (ref)	1.12 (0.52–2.41)	0.775
Model 2 ^b^	1 (ref)	1.13 (0.52–2.46)	0.761
Underweight			
Number of cases, *n* (%)	6 (10.0)	10 (10.3)	
Model 1 ^a^	1 (ref)	1.04 (0.36–3.03)	0.942
Model 2 ^b^	1 (ref)	1.10 (0.37–3.26)	0.865
Missing of household income			
Number of participants, *n*	6	9	
Normal weight			
Number of cases, *n* (%)	4 (66.7)	7 (77.8)	
Model 1 ^a^	1 (ref)	1.26 (0.06–27.46)	0.883
Model 2 ^b^	1 (ref)	1.55 (0.06–39.62)	0.790
Overweight/obesity			
Number of cases, *n* (%)	1 (16.7)	1 (11.1)	
Model 1 ^a^	1(ref)	0.84 (0.03–22.22)	0.917
Model 2 ^b^	1(ref)	0.73 (0.03–19.87)	0.850
Underweight			
Number of cases, *n* (%)	1 (16.7)	1 (11.1)	
Model 1 ^a^	-	-	
Model 2 ^b^	-	-	

^a^ Adjusted for age and sex. ^b^ Adjusted for age, sex, and number of household members.

**Table 5 nutrients-17-00895-t005:** Adjusted means of energy intake according to school lunch groups (*n* = 323).

	School Lunch		** *p* **
	Non-User	User	
Energy intake by body size, kcal			
All ^a^	2235 (2103–2366)	2238 (2173–2389)	0.477
Normal weight	2255 (2092–2417)	2286 (2152–2420)	0.697
Overweight/obesity	2379 (2092–2666)	2366 (2124–2609)	0.920
Underweight	1710 (1304–2116)	1856 (1490–2221)	0.512
Energy intake by body size for boys, kcal			
All ^b^	2640 (2445–2836)	2600 (2441–2759)	0.691
Normal weight	2534 (2308–2760)	2498 (2313–2682)	0.772
Overweight/obesity	3081 (2652–3511)	2816 (2485–3148)	0.219
Underweight	2226 (1319–3134)	2125 (1461–2789)	0.825
Energy intake by body size for girls, kcal			
All ^b^	1695 (1523–1866)	1847 (1702–1993)	0.477
Normal weight	1951 (1647–2255)	2030 (1766–2294)	0.434
Overweight/obesity	1708 (1489–1927)	1921 (1769–2073)	0.084
Underweight	1476 (1172–1780)	1771 (1473–2068)	0.512

^a^ Adjusted for age, sex, household income (<2 million JPY, 2–<6 million JPY, ≥6 million JPY, missing), and number of household members. ^b^ Adjusted for age household income (<2 million JPY, 2–<6 million JPY, ≥6 million JPY, missing), and number of household members.

## Data Availability

National Health and Nutrition Survey data are publicly available from the Ministry of Health, Labour and Welfare, Japan at https://www.mhlw.go.jp/stf/toukei/goriyou/chousahyo.html (accessed on 29 January 2025).

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
