# Peer review of "School Lunch and Body Size in Japanese Junior High School Students: The Japanese National Health and Nutrition Survey"

_nutrients, 2025, doi:10.3390/nu17050895_

Round 1
Reviewer 1 Report
Comments and Suggestions for Authors
General: I think there needs to be revision in the Discussion section. It appears that the authors are taking liberties with the findings
In your abstract and in your results you highlight that there were no associations, but it seems like you are suggesting differences in the Discussion
L22: ...did not support....
The opening paragraph (L30-37) could benefit from revision: 1) delete first sentence, 2) Childhood obesity increases.....; 3) Children who are underweight may also experience.....; 4) a proper diet may help one maintain a normal BMI.
Paragraph 2: Sentence 1: Japan has a national health program... Sentence 2: Children in Japan are....
For L55 and L57 I think its best to cite the authors names and that they were working with Korean populations vs. calling it a Korean study
L66: Avoid 1 sentence paragraphs
L83: if this is a study of middle schoolers, why do we care about 1 year olds?
L115: who is estimating the eyefuls? Eyeful is not a commonly used English word in this case
L118: avoid 1 sentence paragraphs
L127: something seems grammatically off: .....in " I . Household..... Is this correct?
Was a sample size estimation performed (Stat Analysis)
L213: it did not support (remove fully)
L217-232: It seems like the authors are highlighting things not in the tables. There were no associations or any significant p values.
SO you can't say there were gender differences. In fact, I don't think anything in the paragraph is supported by the data.
L234: How were they partially consistent? You had no significant findings. If you have no significant findings then you had nothing in common with the prior study
The rest of the Discussion paragraphs also seem to suffer from this too. If it wasn't statistically significant, you can't suggest any type of difference
Comments on the Quality of English LanguageDon't use 1st person ("We"), change throughout
L30: needs revision for grammar
L45: The lower prevalence of obesity in Japanese children is partially attributed to the national school lunch.....
L55: Bethmann et al evaluated .... in Korean elementary through high school students.....
Author Response
RESPONSE TO REVIEWER 1:
General: I think there needs to be revision in the Discussion section. It appears that the authors are taking liberties with the findings
We sincerely appreciate the reviewers’ efforts in carefully reading our manuscript. In response to your comment, we have revised the manuscript – please see the responses below.
Comment 1:
In your abstract and in your results you highlight that there were no associations, but it seems like you are suggesting differences in the Discussion
L22: ...did not support....
Response: We revised this point as follows:
Line 22: The present findings did not support the expectation for Japanese school lunch pro-gram.
Line 208: As a main result, there was no statistically significant association between school lunch and body size.
Comment 2:
The opening paragraph (L30-37) could benefit from revision: 1) delete first sentence, 2) Childhood obesity increases.....; 3) Children who are underweight may also experience.....; 4) a proper diet may help one maintain a normal BMI.
Response: We revised these points as follows:
Line 30: Childhood obesity increases the risk of obesity in adulthood and increases the risk of chronic diseases such as cardiovascular disease and mortality in young adulthood [1-3]. Additionally, underweight in childhood has been identified as a risk factor for bone fractures in later adulthood [4]. Therefore, maintaining normal weight through a proper diet from childhood is expected to contribute to better health in adulthood. The national health promotion program, “The Health Japan 21 (the third term)”, includes the reduction of overweight and obesity in children and underweight in young women as a goal based on a life-course perspective [5,6].
Comment 3:
Paragraph 2: Sentence 1: Japan has a national health program... Sentence 2: Children in Japan are....
For L55 and L57 I think its best to cite the authors names and that they were working with Korean populations vs. calling it a Korean study
Response: We revised these points as follows:
Line 38: Japan has a national program of school lunch under the law “the School Lunch Program Act” [9]. The percentage of Japanese children who are overweight or obese is relatively low compared to other high-income countries. School lunches have been assumed as a potential reason of this low rate [7,8]. The School Lunch Program Act states that the purpose of school lunches is to “contribute to the sound physical and mental development of children and students”. The Japanese school lunch is provided in accordance with “the School Lunch Nutrition Standards” [10-12]. Previous studies have indicated that higher intakes of protein, dietary fiber, and micronutrients as benefits of school lunches [13,14]. Such nationwide school lunch program is a unique feature of Japan, and is expected to be one of the factors contributing to Japan's lower prevalence of obesity in Japan. One Japanese ecological study (prefectural-level) reported that higher school lunch coverage of junior high school students was associated with lower percentages of those who were overweight or obese especially among boys [8]. How-ever, this previous study reported no difference in the percentage of underweight students or obese girls [8]. To our knowledge, no studies have examined these associations using individual–level data in Japan.
Outside of Japan, there are several research reports that have examined these as-sociations using individual-level data. One Korean study (Kim Y, et al.) reported that elementary through high school students without school lunches tended to have lower odds ratios for obesity than those with school lunches [15]. The other Korean study re-ported that body mass index (BMI) decreased in areas where school lunches were dis-continued compared to areas where school lunches continued, and BMI increased with the resumption of school lunches [16]. In the U.S. studies, there are mixed reports of both increased and decreased rates of obesity among those who participated in school lunch programs, with no unanimous conclusions [17-21]. In summary, studies on the association between school lunches and obesity have had inconsistent conclusions. Furthermore, to our knowledge, the impact of school lunch on childhood underweight in high–income countries has not yet been examined.
Comment 4:
L66: Avoid 1 sentence paragraphs
Response: We revised this point (this sentence was combined to another paragraph):
Comment 5:
L83: if this is a study of middle schoolers, why do we care about 1 year olds?
Response: As you pointed out, this sentence could be misleading and has been removed.
Comment 6:
L115: who is estimating the eyefuls? Eyeful is not a commonly used English word in this case
Response: This was a figurative text and inappropriate. We revised this word as follows:
Line 107: approximate amount.
Comment 7:
L118: avoid 1 sentence paragraphs
Response: We revised this point (this sentence was combined to another paragraph).
Comment 8:
L127: something seems grammatically off: .....in " I . Household..... Is this correct?
Was a sample size estimation performed (Stat Analysis)?
Response: We did not conduct a sample size calculation before the start of this study. As noted in the discussion, not large sample size is one of the limitations of this study.
Comment 9:
L213: it did not support (remove fully)
Response: We deleted this sentence.
Comment 10:
L217-232: It seems like the authors are highlighting things not in the tables. There were no associations or any significant p values.
SO you can't say there were gender differences. In fact, I don't think anything in the paragraph is supported by the data.
Response: With your comments, we revised the relevant discussion descriptions. The revised text is as follows:
Line 210: The Japanese previous study showed that school lunch was negatively associated with overweight/obesity especially among boys [8]. Although the association was not statistically significant in the present study, the odds ratio of overweight/obesity tended to be lower in school lunch users for boys than girls. When energy intake was analyzed by body size in boys only, school lunch users tended to have lower energy in-take than the non-users for overweight/obese individuals. In addition, our finding of the association between school lunch and underweight was not consistent with those of the Japanese previous study [8]. However, since the Japanese previous study was an ecological study using prefecture-level data (47 prefectures) [8], a simple comparison between this previous study and the present study using individual-level data may not be appropriate.
In the present study, the point estimate (the adjusted odds ratio) of underweight was lower than those of overweight/obesity in school lunch users, which can be interpreted as a relatively stronger association. This tendency was particularly pronounced for girls. One potential explanation for this tendency might be the influence of body image concerns and the desire to be thin. A previous study has shown that girls are more likely to desire thinness [28], and it is conceivable that in the absence of school lunch (i.e., when students bring their own lunches or purchase food independently), girls with a stronger desire to be thin may engage in restrictive eating behaviors, leading to an increased risk of underweight. In contrast, school lunches in Japan are pro-vided in accordance with the School Lunch Nutrition Standards, and are uniformly distributed regardless of students’ individual weight-loss intentions. This standardization may help ensure that students, even those with a strong desire to lose weight or who are underweight, consume an adequate amount of food in schools. Supporting this speculation, higher energy intake in the school lunch users compared to the non-users was observed, especially among girls across all body size strata. Because the previous study has suggested that school lunch program reduces income disparities in micronutrient insufficiency among elementary school students [14], school lunch pro-gram may have a favorable role in addressing nutritional insufficiency relatively.
Comment 11:
L234: How were they partially consistent? You had no significant findings. If you have no significant findings then you had nothing in common with the prior study
The rest of the Discussion paragraphs also seem to suffer from this too. If it wasn't statistically significant, you can't suggest any type of difference
Response: As per our response to Comment 10, we revised the relevant discussion descriptions.
Line 208: As a main result, there was no statistically significant association between school lunch and body size.
The Japanese previous study showed that school lunch was negatively associ-ated with overweight/obesity especially among boys [8]. Although the association was not statistically significant in the present study, the odds ratio of overweight/obesity tended to be lower in school lunch users for boys than girls.
Comment 12:
Comments on the Quality of English Language
Don't use 1st person ("We"), change throughout
L30: needs revision for grammar
L45: The lower prevalence of obesity in Japanese children is partially attributed to the national school lunch.....
L55: Bethmann et al evaluated .... in Korean elementary through high school students.....
Response: As you indicated, we conducted proofreading the entire text of the manuscript.
Reviewer 2 Report
Comments and Suggestions for Authors
NUTRIENTS-3477925 presents findings for school lunch programs and body size in Japanese youth. While some parts of the paper were interesting, other areas could be improved. I hope the authors consider my feedback.
· Line 30: Starting sentences with “It” should be avoided here and elsewhere.
· Lines 69-72: Given the inconsistency in the research reported in the Intro, maybe just delete this hypothesis sentence. Revision will also be needed at lines 214-216.
· Figure 1 should be more consistent in reporting. For example, in the upper box, total n= and n= by year are listed. Exclusions are then based on specific missing data. The lower box includes n= overall and by gender. Please revise for consistency in some regard.
· Section 2.4 and body size section could be merged? Same could apply to Section 2.3 and Energy Intake?
· Section 2.6: Consent statement of sort is needed.
· Lines 135-137: Model 1 could be stripped to be cruder by removing age and gender. Then add all the covariates (including age and gender) to Model 2 as the fully-adjusted model.
· Lines 138-143: Need to list the implications of sensitivity analyses. For example, sensitivity analyses test against the main findings for factors such as bias from missing data. Otherwise, these analyses are more so supplementary.
· Line 220 and elsewhere: Remove results from a Discussion section.
· Line 221 and elsewhere: Avoid re-presenting data elements (e.g., Tables) for disruption of prospective reader flow.
· Discussion: There needs to be some acknowledgement of the political policy implications of this research, including other possible benefits of the school lunch program outside of body size. The paper in current form has a negative slant in this regard, and softening may help.
· Make any changes to the abstract that align with those in the text.
Author Response
RESPONSE TO REVIEWER 2:
NUTRIENTS-3477925 presents findings for school lunch programs and body size in Japanese youth. While some parts of the paper were interesting, other areas could be improved. I hope the authors consider my feedback.
We sincerely appreciate the reviewers’ efforts in carefully reading our manuscript. In response to your comment, we have revised the manuscript – please see the responses below.
Comment 1:
- Line 30: Starting sentences with “It” should be avoided here and elsewhere.
Response: Considering the comments of other reviewers, this sentence was removed.
Comment 2:
- Lines 69-72: Given the inconsistency in the research reported in the Intro, maybe just delete this hypothesis sentence. Revision will also be needed at lines 214-216.
Response: According this comment, we removed this sentence.
Comment 3:
- Figure 1 should be more consistent in reporting. For example, in the upper box, total n= and n= by year are listed. Exclusions are then based on specific missing data. The lower box includes n= overall and by gender. Please revise for consistency in some regard.
Response: We revised Figure 1 according this comment:
Line 86: Figure 1. Flowchart of study participants.
Comment 4:
- Section 2.4 and body size section could be merged? Same could apply to Section 2.3 and Energy Intake?
Response: We combined these sections.
Comment 5:
- Section 2.6: Consent statement of sort is needed.
Response: We added the following text to the manuscript to address these points:
Line 122: The National Health and Nutrition Examination Survey is conducted by the Japanese government under the Japanese law “Health Promotion Act”, and research utilization of Japanese government statistics is stipulated in the “Statistics Act”.
Comment 6:
- Lines 135-137: Model 1 could be stripped to be cruder by removing age and gender. Then add all the covariates (including age and gender) to Model 2 as the fully-adjusted model.
Response: We revised these sentences as follows:
Line 129: Adjustment items were gender, age, household income, and number of household members.
Comment 7:
- Lines 138-143: Need to list the implications of sensitivity analyses. For example, sensitivity analyses test against the main findings for factors such as bias from missing data. Otherwise, these analyses are more so supplementary.
Response: We added the following text to the manuscript to address these points:
Line 131: Three types of sensitivity analyses were also performed. First, because the main analysis included overweight to account for the situation that obesity is relatively rare in the Japanese population, we performed an analysis with obesity (not including overweight) as the outcome variable. Second, to check whether the results for each gender are consistent with the main results, a stratified analysis by gender was performed. Third, to examine whether the impact of school lunch on body size differs depending on household income (e.g., positive association between school lunch and normal weight is stronger for students with lower household income), a stratified analysis by household income was performed.
Comment 8:
- Line 220 and elsewhere: Remove results from a Discussion section.
Response: We deleted results (values) from the Discussion section.
Comment 9:
- Line 221 and elsewhere: Avoid representing data elements (e.g., Tables) for disruption of prospective reader flow.
Response: With your comments, we deleted data elements (e.g., Tables).
Comment 10:
- Discussion: There needs to be some acknowledgement of the political policy implications of this research, including other possible benefits of the school lunch program outside of body size. The paper in current form has a negative slant in this regard, and softening may help.
Response: We added the following text to the manuscript to address these points:
Line 234: Because the previous study has suggested that school lunch program reduces income disparities in micronutrient insufficiency among elementary school students [14], school lunch program may have a favorable role in addressing nutritional insufficiency relatively.
Comment 11:
- Make any changes to the abstract that align with those in the text.
Response: We revised sentences in the abstract.
Round 2
Reviewer 1 Report
Comments and Suggestions for Authors
L53: would recommend deleting the Kim Y et al
L106: (e.g., whole meal). List what is important, don’t use etc.
L116: try to avoid 1 sentence paragraphs
L209: The Japanese previous study (is not grammatically correct)
A prior study by __________, based out of Japan, showed that school….
L211: if something isn’t statistically significant how can you make a claim about something being lower
L211-217: I think its hard to draw comparisons between studies
L219: but its not stat significant right? So how do you know it is actually less?
Author Response
RESPONSE TO REVIEWER 1:
We sincerely appreciate the reviewers’ efforts in carefully reading our manuscript. In response to your comment, we have revised the manuscript – please see the responses below.
Comment 1:
L53: would recommend deleting the Kim Y et al.
Response: We deleted this description.
Comment 2:
L106: (e.g., whole meal). List what is important, don’t use etc.
Response: We revised these points as follows:
Line 104: For the school lunches, the list of foods used for meal planning was obtained and participants were asked leftover, and nutritionist researchers calculated their intake [27].
Comment 3:
L116: try to avoid 1 sentence paragraphs
Response: We revised these points.
Comment 4:
L209: The Japanese previous study (is not grammatically correct)
A prior study by __________, based out of Japan, showed that school….
L211: if something isn’t statistically significant how can you make a claim about something being lower
L211-217: I think its hard to draw comparisons between studies
L219: but its not stat significant right? So how do you know it is actually less?
Response: In light of your series of comments on this paragraph, we revised this part as follows:
Line 206: The previous study has shown that school lunch was negatively associated with overweight/obesity, especially among boys [8]. On the other hand, in the present study, the association was not statistically significant, regardless of gender. However, since this previous study was an ecological study using prefecture-level data (47 prefectures in Japan) [8], a simple comparison may not be appropriate. Future research at the individual level will be required to obtain integrative findings such as meta-analysis.
Line 230: However, since none of the results in the present study were statistically significant, the above speculations should be confirmed by future studies.
Reviewer 2 Report
Comments and Suggestions for Authors
Additional feedback for the authors:
***There needs to be more text devoted to balance in policy implications related to this paper. A single sentence related to disparities in micronutrients is inadequate. Please devote more text (paragraph) that fairly discusses the advantages and disadvantages of such programs in Japan.
***The conclusion that individualized nutrition management is needed does not align with your study. This is more so a future research implication. Please edit to make this statement a future research implication, and revise the Conclusions in the Abstract and main text to better reflect the findings of the study.
Author Response
RESPONSE TO REVIEWER 2:
We sincerely appreciate the reviewers’ efforts in carefully reading our manuscript. In response to your comment, we have revised the manuscript – please see the responses below.
Comment 1:
***There needs to be more text devoted to balance in policy implications related to this paper. A single sentence related to disparities in micronutrients is inadequate. Please devote more text (paragraph) that fairly discusses the advantages and disadvantages of such programs in Japan.
Response: We added the following text to the manuscript to address these points:
Line 258: There are both advantages and disadvantages to implementing the uniform school lunch approach, such as the program in Japan. As an advantage, it has been reported that the school lunch may reduce the gap in micronutrient intake even among children from different socioeconomic backgrounds, as aforementioned [14]. These expectations have been suggested worldwide [31-34]. Japanese school lunch is a nutrition policy that is equally accessible to all children [33]. The current Japanese school lunch program has a historical background for students who were starvation after World War II, and nutrient sufficiency has been a target of the school lunch program for more than half a century [12]. However, in Japan since the 21st century, it has been reported that the frequency of child obesity is higher in households with lower economic status, and it is difficult to say that starvation is more common public health issue even among households with lower economic status [35-37]. Additionally, the uniformity of the meal content emphasizes the educational value of teaching children about food and food culture in relation to nutrition [38]. On the other hand, one disadvantage is that the uniform school lunch approach may not address individual dietary needs and preferences. Preferences have been reported as factors associated with leftover school lunches in Japanese junior high schools [29]. Using the example of school lunches in Japan and France, the previous study has also suggested that there is concern that education that encourages uniform school lunches may result in children whose school lunch content does not match their personal culture and preferences, thereby contributing to social exclusion [33,39]. Furthermore, for strategies to increase school lunch consumption, research evidence also supports offering students more menu choices and adapting recipes to improve the palatability and/or cultural appropriateness of foods [40]. A recent systematic review stated that “evidence shows voluntary policies are generally not an effective approach for shaping and protecting a healthy food environment” regarding the school lunch program in Japan [41]. This previous study concluded that “Future research should evaluate the implementation and efficacy of national school meal policies for improving health” [41].
Comment 2:
***The conclusion that individualized nutrition management is needed does not align with your study. This is more so a future research implication. Please edit to make this statement a future research implication, and revise the Conclusions in the Abstract and main text to better reflect the findings of the study.
Response: We revised these points as follows:
Line 320: Additionally, the findings of the present study did not clarify whether the school lunch program contributes to providing an appropriate amount of meal for each student. Further study would be required to implement a school lunch program that takes into account individualized nutrition management to provide an appropriate serving size for each student.